# Eco-Efficiency Analysis for the Russian Cities along the Northern Sea Route: A Data Envelopment Analysis Approach Using an Epsilon-Based Measure Model

**DOI:** 10.3390/ijerph18116097

**Published:** 2021-06-05

**Authors:** Shuaiyu Yao, Mengmeng Chen, Dmitri Muravev, Wendi Ouyang

**Affiliations:** 1School of Naval Architecture, Ocean and Civil Engineering, Shanghai Jiao Tong University, Shanghai 200240, China; shuaiyuyao@sjtu.edu.cn (S.Y.); dmitri_muravev@sjtu.edu.cn (D.M.); 2School of Accounting, Guangzhou Huashang College, Guangzhou 510000, China; 3Department of Logistics and Transportation Systems Management, Mining Engineering and Transport Institute, Nosov Magnitogorsk State Technical University, 455000 Magnitogorsk, Russia; 4College of Economics and Management, Nanjing University of Aeronautics and Astronautics, Nanjing 211100, China; wendi.ouyang@nuaa.edu.cn

**Keywords:** eco-efficiency, Northern Sea Route, Russian cities, epsilon-based measure, data envelopment analysis

## Abstract

In this paper, an eco-efficiency analysis is conducted using the epsilon-based measure data envelopment analysis (EBM-DEA) model for Russian cities along the Northern Sea Route (NSR). The EBM-DEA model includes five input variables: population, capital, public investment, water supply, and energy supply and four output variables: gross regional product (GRP), greenhouse gas (GHG) emissions, solid waste, and water pollution. The pattern of eco-efficiency of 28 Russian cities along the NSR is empirically analyzed based on the associated real data across the years from 2010 to 2019. The empirical results obtained from the analysis show that St. Petersburg, Provideniya, Nadym, N. Urengoy, and Noyabrsk are eco-efficient throughout the 10 years. The results also indicate that the cities along the central section of the NSR are generally more eco-efficient than those along other sections, and the cities with higher level of GRPs per capita have relatively higher eco-efficiency with a few exceptions. The study provides deeper insights into the causes of disparity in eco-efficiency, and gives further implications on eco-efficiency improvement strategies. The contributions of this study lie in the fact that new variables are taken into account and new modeling techniques are employed for the assessment of the eco-efficiency of the Russian cities.

## 1. Introduction

Russia is one of the most significant stakeholders in the Artic region. The Russian Arctic covers an area of about 3 million m^2^ (18% of the Russian territory), including 2.2 million m^2^ of land [1]. It has a population of about 2.4 million, about 1.5% of the Russian population, but generates around 10% of Russia’s gross domestic product (GDP) [2]. The Russian Arctic contains vast deposits of natural resources, in particular, petroleum [3,4]. The total amount of undiscovered petroleum in the Arctic area has been estimated to be 413 BBOE (billion barrels of oil equivalent), which accounts for about 22% of the world’s undiscovered conventional oil and gas resources [5]. In terms of oil and gas, Russia has a larger share than other Arctic countries, with its oil accounting for 41% and its natural gas for about 70% of the total Arctic resources [5]. In addition, large quantities of Russian proven mineral resources are located in the Arctic, such as more than 96% of Russian platinum metals, over 90% of nickel and cobalt, and about 60% of copper [3].

For decades, Russia has been a major producer and exporter of natural resources and its economy growth is driven by the associated exports [6,7,8]. The natural resources in the Russian Arctic can be exported from the ports along the Northern Sea Route (NSR). The NSR in the Arctic Ocean provides a shortcut between Europe and Asia [9]. Compared with the use of Suez Canal route, the use of NSR can reduce the shipping distance between Europe and Asia by roughly 5000 nautical miles [10], and thus can help shipping companies save tremendously on logistics costs [11]. The speed of the Arctic ice meltdown is forecasted to be more dramatic than previously predicted, and the travel time of NSR is increasingly shorter [12]. The opening up of the NSR presents a new area of exploration for natural resources [13]. In addition, the NSR is far away from unsafe areas, such as the Middle East and the North Africa. Due to the above facts, the maximum volume of cargo flows between Asia and Europe is expected to be as large as 46 million 20-feet equivalent unit [11], and the NSR potentially have a positive influence on the related stakeholders’ benefits [14].

However, it will be difficult for a single country to develop the NSR and the natural resources in the Russian Arctic. Its development will require extensive human resources, advanced technologies and equipment, as well as high quantities of capital investment [15]. Hence, the enhancement of regional cooperation along the NSR will contribute to a win–win situation for all the related stakeholders. Russia has introduced preferential tax policies to attract international investment to exploit natural resources. State-owned Rosneft has formed joint ventures with international companies such as Exxon, Statoil, and Eni. The Yamal liquefied natural gas (LNG) project, the Payaha gas field project, and the Kupol gold mine are typical examples of international cooperative development projects [8].

For a long time, the Russian authorities have prioritized resource-based economic development over environmental issues, which has led to environmental damage. As an environmentally fragile area, the Russian Arctic is at great risk due to overexploitation and the activities involved in using the NSR [4,16]. It is estimated that as much as 4 million tons of industrial and construction debris, and 4–12 million barrels of steel are lying along the Arctic Ocean coast [17]. Environmental damage in the Arctic is also a legacy of history. With the collapse of the Soviet Union, polar explorers left the Arctic, leaving behind buildings, cars, unused fuel, spare parts, and building materials. In addition, mines of gold, tin, and mercury shut down, leaving vast deposits of rock and slag [17]. With climate warming, the accelerating loss of the Arctic sea ice brings negative effects on the Arctic and even global ecosystems [11,15]. The shrinkage of Arctic sea ice threatens some species (e.g., polar bears and seals) that depend on sea ice [12,15]. Meanwhile, the habitats of Arctic fish and plankton disappear due to the sea ice shrinkage and the increase in water temperatures [18]. Another significant impact of climate change on the Russian Arctic is the degradation of permafrost, which has a negative impact on the structural integrity of infrastructure [19]. Furthermore, the degradation of permafrost may affect biochemical process enhancing leaching and migration of trace metals in permafrost-affected soils, which can have negative influence on Arctic ecosystem [20].

Facing the fragile ecological environment in the Russian Arctic, Russian government is taking measures to protect it and facilitate regional sustainable development. For example, the “Arctic Strategic Action Program 2009” proposes measures to prevent, eliminate, and reduce the consequences of adverse environmental impacts. In 2010, Putin launched a plan to clean up the Russian Arctic. A waste disposal practice began in 2012 and takes place every summer in the polar islands of Barents Sea and other Arctic regions [4]. In 2012, Putin approved the strategy document of the “State Environmental Policy for the period up to 2030” (Order of the President 2012) to establish a mechanism for implementing the environmental protection. Another active measure was the creation of a new legal order for national parks and nature reserves in the Russian Arctic. Under such an order, a national park and state nature reserves was established [16]. Meanwhile, the Russian Federation submitted the Intended Nationally Determined Contribution proposal to the United Nations, which is aimed at reducing net greenhouse gas (GHG) emissions by 25–30% by 2030, compared to the levels observed in 1990 [16].

The economic development of the Russian Arctic is achieved at the cost of environmental pollution and ecological destruction. Excessive environmental protection can put a brake on economic development in some cases. Such situations drive the Russian government to balance economic development and environmental protection to ensure sustainable development of the Russian Arctic. It is hence of great significance to evaluate and analyze the relationship between the economic development and environmental protection of the Russian Arctic. Eco-efficiency can be appropriate for measuring the relationship between economic development and environmental protection [21,22].

In this study, data envelopment analysis (DEA) is used to evaluate the eco-efficiency of the Russian cities along the NSR (most of them are located in the Russian Arctic), as it has been widely used in the areas regarding evaluation of economic and environmental sustainability [23]. The pattern of eco-efficiency of these cities is empirically analyzed, and the study provides further insights into the analysis results and eco-efficiency improvement strategies.

The major contributions of this study can be summarized in three aspects as follows.

Comprehensive panel data for the period of 2010–2019 (shown in Appendix A) are collected for the evaluation and analysis of the eco-efficiency of the Russian cities along the NSR. The panel data can establish a sound foundation for further studies.

Compared with previous studies, this study introduces new variables in the evaluation of ecological total-factor energy efficiency (TFEE). It can provide a new perspective for the related stakeholders to recognize, evaluate, and analyze more comprehensively and deeply the eco-efficiency of the Russian cities along the NSR.

The epsilon-based measure data envelopment analysis (EBM-DEA) model is applied to the evaluation and analysis of sustainable development of the Russian cities along the NSR. The EBM-DEA model can effectively solve the problems, which exist in radial and non-radial direction.

The remainder of this paper is organized as follows. Section 2 reviews literatures on the definition, evaluation methods, and applications of eco-efficiency. The EBM-DEA methodology is introduced in Section 3. Section 4 presents the used data, data processing techniques, and statistical analysis of this study. Section 5 provides a presentation of results and associated discussion. Some conclusions are summarized in Section 6.

## 2. Literature Review

### 2.1. Definition of Eco-Efficiency

In the traditional sense, eco-efficiency is a term used to describe the quantity of economic benefits per unit of ecological energy [24]. Higher eco-efficiency requires a country to generate more economic output with a lower cost of ecological resources. However, eco-efficiency has been given different meanings [25]. The Business Council for Sustainable Development defined it as being achieved by providing a competitively priced product or service, which satisfies a high standard of living such that negative impact of economic development on the environment is at a tolerable level throughout the life cycle [26]. The Organisation for Economic Cooperation and Development (OECD) defined it as the efficiency with which ecological resources are used to meet human needs [27]. It can be considered as a ratio of an output divided by an input [27]. This definition extends the application of eco-efficiency to governments, industries, and other sectors from the perspective of input and output [27]. Although there are various definitions of eco-efficiency, the overall target of being eco-efficient is to obtain the maximum economic benefit with the minimum cost of environment and ecology.

### 2.2. Eco-Efficiency and DEA

Eco-efficiency is currently a research focus due to its theoretical value and practical significance [28,29]. It has been applied to a wide variety of industrial and regional contexts [25]. There are various methods for eco-efficiency evaluation, which include life cycle analysis [30,31], ecological footprint [32,33], energy analysis [34,35], and ratio method [36]. DEA is also a major method of evaluating eco-efficiency, which takes into account economic benefits and ecological performance. Hailu and Veeman [37] used DEA to analyze the eco-efficiency of the Canadian paper industry, and they proposed a non-parametric analysis method to incorporate undesirable or pollutant output into productivity growth. The DEA method is applied by Wursthorn et al. [38] to perform the analysis of environment-economic trade-off and eco-efficiency of industrial processes. Wang et al. [39] took Xinfa eco-industrial parks as a case study and developed a matrix network of DEA model to evaluate ecological industry chain efficiency, which takes into account energy, economic, and environmental constraints.

Many DEA studies have also focused on regional or international eco-efficiency. Zhou et al. [40] proposed two slack-based efficiency measures for modeling of environmental performance of 30 OECD countries, and four variables, i.e., energy supply, population, GDP, and carbon dioxide (CO_2_) emission are included in the associated model. Li and Hu [41] constructed slacks-based measure data envelopment analysis (SBM-DEA) models to calculate the ecological TFEE of 30 regions in China from 2005 to 2009. These models take total energy consumption, total capital stock, and total labor force as inputs and GDP, CO_2_, and sulfur dioxide (SO_2_) as outputs. Zhang et al. [42] used the SBM-DEA models to calculate the ecological TFEE of 30 provinces of China. Based on CO_2_ and SO_2_ emissions and chemical oxygen demand (COD) in China from 2001 to 2010, they carried out an empirical analysis of regional ecological energy efficiency. Based on the SBM-DEA models, Choi et al. [43] analyzed the efficiency of CO_2_ emission and energy, potential CO_2_ emission reduction, and marginal cost of CO_2_ emission in 30 provinces of China from 2001 to 2010. Li et al. used DEA to study the eco-efficiencies of China at provincial levels and the associated driving factors [44]. Lorenzo-Toja et al. [45] extensively analyzed 113 wastewater treatment plants across Spain using the methodology that combines life cycle assessment (LCA) and DEA to determine the operational efficiency of each plant in order to obtain environmental benchmarks for inefficient plants. Halkos and Petrou [29] studied the eco-efficiency of the 28 EU countries in 2008, 2010, 2012, and 2014, which uses DEA and directional distance functions to deal with undesired outcomes.

It is evident from the above studies that DEA has been widely used to evaluate eco-efficiency. In addition, DEA has been proven to be a useful and valuable tool for decision makers. Hence, DEA was used in this study to evaluate the eco-efficiency of environmental governance and economic development of Russian cities along the NSR.

### 2.3. DEA Models

Compared with other methods, DEA has the following advantages. (1) It does not require the estimation of the production function in advance [43]. (2) It gives objective weights to different environmental factors based on data and does not depend on human judgment [39]. (3) It can describe the effective production frontier and provide a benchmark for the efficiency improvement of invalid decision-making units (DMUs) [46]. (4) It explains multi-input and multi-output systems for efficiency measurement [47]. Based on these properties, non-parametric frontier analysis represented by DEA has been widely applied to efficiency measurement due to its unique flexibility and applicability [48].

The DEA model mainly evaluates the relative efficiency of DMUs. It generates the efficiency by analyzing the frontier of input and output variables. It has several variants, e.g., the Charnes–Cooper–Rhodes (CCR) model, the Banker-Charnes-Cooper (BCC) model, the SBM model, and the EBM model. The CCR model proposed by Charnes et al. [47] assumes that the return on scale is constant, but there is rarely a constant return on scale in the real world. Banker et al. [49] proposed the BCC model by extending the CCR model. The BCC model assumes a scale return for variables. The BCC model accepts scale return in constant or decreasing marginal productivity. The CCR and BCC are radial models when all of their input and output variables change in the same proportion. The conventional models do not consider the non-radial slacks, so the results ignore some inefficiency impacts. The SBM model proposed by Tone [50] is a non-radial model. It adds slacks into the objective function, which deals with the problem of undesired output. Compared with the CCR and BCC models, the SBM model averts the deviation and influence caused by radial and angle differences, which can better reflect the essence of efficiency evaluation. Although the SBM model can describe all slacks information, it ignores the overall proportional changes of variables. Another issue comes from the SBM model’s property, i.e., linear programming, where the optimal loose case presents a strong contrast between positive and zero values [51]. This leads to an underestimation that is inconsistent with the actual situation.

The CCR and BCC model are both radial DEA models, where non-radial relaxation variables are ignored. Moreover, the SBM model fails to consider the characteristics of the radial model. Tone and Tsutsui [51] proposed the EBM model, which integrates radial and non-radial features in a unified framework. It reflects the difference between the optimal observed value and the real value. In addition, the EBM model takes slacks into consideration to reflect the difference between the non-radial parts of inputs and outputs. The results of the EBM model consider the framework of both the CCR and SBM model, and it can thereby calculate the efficiency of DMU more accurately.

## 3. Methodology

Suppose that there are n DMUs in this study. Each DMU denoted by $DMU_{j}$ (j = 1, …, n) has m inputs (i = 1, …, m) and s outputs (r = 1, …, s). The input and output of $DMU_{j}$ are denoted by $X=\left\{x_{ij}\right\}\in R^{m\times n}$ and $Y=\left\{y_{ij}\right\}\in R^{s\times n}$, respectively. It is assumed that X>0 and Y>0. Based on the terminology introduced above, the CCR, SBM, and EBM model are briefly introduced in the following part of this section.

### 3.1. CCR Model

Under the assumption of constant returns to scale, the input-oriented CCR model is used to evaluate the technical efficiency $\theta^{*}$ of DMU based on the following linear optimization program.
(1)$$\begin{array}{ll} \theta^{*}= & \underset{\theta ,\lambda ,s^{-}}{\min}\;\theta \\ & subject\;to\;\left\{ \begin{array}{l} \sum_{j=1}^{n}x_{ij}\lambda_{j}+s^{-}=\theta X_{0},\;i=1,\ldots ,m\\ \sum_{j=1}^{n}x_{rj}\lambda_{j}+s^{+}=Y_{0},\;i=1,\ldots ,s\\ s^{-},s^{+},\lambda_{j}\geq 0,\;j=1,\ldots ,n \end{array}\right. \end{array}$$

In Equation (1), the range of θ is 0≤θ≤1. $s^{-}$ and $s^{+}$ represent the non-radial slacks of each input and output of DMUs, respectively. λ indicates the intensity vector.

### 3.2. SBM Model

Under the assumption of constant returns to scale, the input-oriented SBM model proposed by Tone [50] can be used to evaluate the efficiency $\tau^{*}$ of DMU using the linear optimization program shown in Equation (2).
(2)$$\begin{array}{ll} \tau^{*}= & \min\left(1-\frac{1}{m}\sum_{i=1}^{m}\frac{s_{i}^{-}}{x_{i0}}\right)\\ & subject\;to\;\left\{ \begin{array}{l} x_{i0}=\sum_{j=1}^{n}x_{ij}\lambda_{j}+s_{i}^{-},\;i=1,\ldots ,m\\ y_{i0}\leq \sum_{j=1}^{n}y_{ij}\lambda_{j},\;i=1,\ldots ,s\\ \lambda_{j}\geq 0\left(\forall j\right),\;s_{i}^{-}\geq 0\left(\forall i\right) \end{array}\right. \end{array}$$

In Equation (2), λ denotes the intensity vector, and $s^{-}={\left(s_{1}^{-},\ldots ,s_{m}^{-}\right)}^{T}$ represents the non-radial input slacks vector.

### 3.3. EBM Model

The EBM model proposed by Tone and Tsutsui [51] has both radial and non-radial features in a unified framework. The objective function of EBM is shown as follows:(3)$$\begin{array}{ll} \gamma^{*}= & \underset{\theta ,\lambda ,s^{-}}{\min}\theta -\varepsilon_{x}\sum_{i=1}^{m}\left(\frac{w_{i}^{-}s_{i}^{-}}{x_{i0}}\right)\\ & subject\;to\;\left\{ \begin{array}{l} \theta x_{0}-X\lambda -s^{-}=0\\ Y\lambda \geq 0\\ \lambda \geq 0\\ s^{-}\geq 0 \end{array}\right. \end{array}$$

In Equation (3), $\gamma^{*}$ is the optimal efficiency score of EBM model. θ is the radial efficiency value calculated by the CCR model. λ represents the weight vector. $w_{i}^{-}$ is the weight (relative importance) of input i and satisfies $\sum_{i=1}^{m}w_{i}^{-}=1\;\left(w_{i}^{-}\geq 0,\;\forall i\right),\;where\;w_{i}^{-}$ should be provided prior to efficiency measurements. $\varepsilon_{x}$ combines the radial θ and non-radial slacks. The properties and related definitions of EBM model are shown as follows.

**Proposition** **1.**
*If*
θ
*= 1 and*
ε
*= 1, the model simplifies to an input-oriented SBM model.*


**Proposition** **2.**
*The model has a finite optimal value,*
$\varepsilon_{x}\epsilon \left[0,1\right]$
*.*


**Proposition** **3.**
$\gamma^{*}$
*is non-increasing in*
$\varepsilon_{x}$
*.*


**Definition** **1.***(EBM input-efficiency). When*$\gamma^{*}=1$, *DMU_0_ is called EBM input efficiency.*

**Definition** **2.***(EBM projection). Let the optimal solution to Equations (1)–(3) be* ($\theta^{*},\lambda^{*},s^{-*}$). *Tone and Tsutsui (2010) defined the projection of DMU*
$\left(x_{0},y_{0}\right)$
*as follows:*
(4)$$\begin{array}{l} x_{0}^{*}=X\lambda^{*}=\theta^{*}x_{0}-s^{-*},\\ y_{0}^{*}=Y\lambda^{*}. \end{array}$$

## 4. Data Preparation and Model Framework

EBM models are used for the eco-efficiency analysis of the 28 Russian cities along the NSR in this study. The variables used in the models include population, capital, public investment, water supply, energy supply, gross regional product (GRP), GHG emissions, solid waste, and water pollution. The first five of the abovementioned variables are used as the inputs of the EBM models, and the rest are taken as the associated outputs. Figure 1 exhibits such a model framework. These variables are measured in the units displayed in Table 1. Annual data on these variables for 28 Russian cities along the NSR in the period from 2010 to 2019 are obtained from the Russian Federal Statistics Service. These data are used as inputs and outputs of the models constructed in this study. The descriptive statistics of these inputs and outputs for all the years are exhibited in Table 2.

## 5. Results and Discussion

The eco-efficiency scores of the cities along the NSR are generated based on the EBM model, which are exhibited in Table 3. As shown in Table 3, St. Petersburg, Provideniya, Nadym, N. Urengoy, and Noyabrsk are eco-efficient across all the years from 2010 to 2019. Onega is moderately eco-efficient, and the eco-efficiency scores range between 0.345 and 0.420. The other cities in Table 3 have relatively low levels of eco-efficiency. From the perspective of the temporal pattern of eco-efficiency scores (those of the cities along the western, central, and eastern section of the NSR are displayed in Figure 2, Figure 3 and Figure 4, respectively), it is evident that the scores of most cities remain stable or fluctuate slightly around a certain level. The exceptions are Naryan-Mar, Novodvinsk, and Salekhard, which all experience steady rise and sharp decline.

Regarding the spatial distribution of the eco-efficiency of these cities (shown in Figure 5), it can be seen that the cities along the central section of the NSR are generally more eco-efficient than those along the eastern and western section. This regional disparity in eco-efficiency exhibits a similar pattern to the GRPs per capita of the cities. As can be seen from Figure 6, the cities with higher GRPs per capita also have higher eco-efficiency scores with the exception of St. Petersburg and Provideniya (denoted by “1” and “8”, respectively). A possible explanation for the association is based on natural resources, populations, and policies. The neighboring areas of the central section of the NSR containing rich reserve of natural resources such as oil and gas can generate relatively high GRPs, and the areas have small populations, which leads to comparatively high GRPs per capita. In addition, the Russian economy relies heavily on the export of natural resources, and the Russian government provides a strong support for the development of the energy industry. This all indicates that the authorities and related stakeholders can invest more resources in the fields such as capital, technology, and management to the cities along the eastern and western section, so as to make these cities use resources more efficiently and discharge fewer pollutants.

To look at how the eco-efficiency of the cities might be improved, it is necessary to generate the results of inputs and outputs optimization. The average annual percentage changes in the inputs and outputs for these cities are shown in Table 4. The “S-”, “S+”, and “SB” in Table 4 indicate the excesses of inputs, shortfalls of positive outputs, and excesses of negative outputs, respectively, according to the efficiency (optimal solution) achieved by the EBM-DEA models. It is clear from Table 4 that 18 out of 28 cities need to have a more than 50% reduction in both capital and public investment to achieve eco-efficiency. This leads to the inefficiency in capital and public investment of many Russian Arctic cities. In fact, this does not mean that capital and public investment in these cities need to be reduced. It is worth noting that the relative efficiency benchmarks (e.g., St. Petersburg) in these cities are insufficient in their use of capital and public investment. It is recommended to diversify the local industrial structure and extend the economic functions, which is one of the ways for inefficiency to be resolved in the Russian cities along the NSR. In addition to the conventional industries such as oil, gas, and mining industries, the authorities can vigorously develop the fishery economy based on the considerable fishery resource in the Russian Arctic. Arctic tourism can be a promising sector of the Russian economy, which has a multiplicative effect for the development of the infrastructure, social services, and employment in the Russian cities along the NSR. In addition, support tools should be used to attract investment to these Russian cities. These tools include lower profit tax rates; reduced severance tax coefficients for oil, gas, and mineral development; a notifying procedure for value-added tax refunds; a simplified procedure for land plots supply; and invariable terms for investment projects implementation.

In terms of population, some cities such as Onega, Vladivostok, and Olenegorsk need to reduce their populations significantly to achieve eco-efficiency according to the results of EBM-DEA models. The authorities need to develop a set of preferential policies to encourage immigration between the associated cities, so that the cities that have a real demand of population can obtain population growth and those that have excessive populations can obtain fewer population to achieve eco-efficiency. Regarding water supply, the cities such as Murmansk, Kandalaksha, Dudinka, Vanino, Nakhodka, Norilsk, and Olenegorsk need to take measures to reduce the water supply substantially to achieve eco-efficiency. With regards to energy supply, in order to be eco-efficient, 10 out of 28 cities need to cut their energy supply by more than 40%. In respect of GRP, a remarkable fact is that the GRPs of most cities need to be increased substantially (the cities, e.g., Dudinka, Vorkuta, Revda, and Nikel even need to have an increase of more than 1000% on GRPs), so that these cities can achieve eco-efficiency. This suggests that according to the efficiency generated by the models, most of the Russian cities along the NSR are economically inefficient, which is generally attributed to the factors such as outdated infrastructure, tiny populations, harsh natural environment, simple economic structure, and insufficient financial and technological resources. It is recommended to update these cities’ outdated infrastructure, and develop more regional infrastructure and transportation projects. The “North Latitude Passage” is one of the related key infrastructure projects. It will advance the effective development of the rich natural resources in the Russian Arctic areas (e.g., Polar Urals, Yamal, and the north of Krasnoyarsk territory). The authorities need to continue developing the communication and coastal infrastructure (e.g., navigational and hydrometeorological aids, and port facilities) along the NSR, which will ensure safety of commercial transits through the NSR. The successful functioning of the NSR will bring more development opportunities to the cities along the NSR. The logistical hubs at the end points of the NSR (e.g., the ports of Murmansk and Petropavlovsk-Kamchatsky) can be created to serve domestic and international shipping. The Russian Arctic has great potential for economic development due to its rich natural resources and geographical significance. Its infrastructure construction and oil, gas, and mineral resource development have become the main contributors to current economic growth [53]. It is suggested to promote the further construction and development of the related projects (e.g., the projects of Prirazlomnoye oil field, Novy Port oil field, Bovanenkovo gas field, Kharasaveyskoye gas field, Yamal LNG, and Arctic LNG 2), which can potentially help these cities attract more residents and business opportunities, so that the economy of these cities can be stimulated and boosted.

Regarding the excesses of negative outputs (denoted by “SB” in Table 4), including GHG, solid waste, and water pollution, more than half of the cities need to have a reduction of more than 40% on these outputs to achieve eco-efficiency. The ecology of the Russian Arctic is fragile, and the impact of environmental damage is greater than that in other regions. The circular economy will become the sustainable development solution for Russian Arctic cities in the future. It can reduce the emissions of various forms of pollution and waste, and ensure the sustainable development of the region as much as possible [54]. To maintain the balance between economic development and Arctic environmental protection, the authorities need to continue conducting a major clean-up of the environmental damage in the Russian Arctic areas which was accumulated through the economic activities in the past decades. They also need to develop a system of specially protected natural territories and reserves in the Russian Arctic for better environmental protection. Education and science centers need to be established in the Russian cities along the NSR to ensure the development of fundamental research and help address the practical tasks of Arctic sustainable development. International research teams and alliances of high-tech companies should be encouraged to take part in joint research projects in the fields such as shipbuilding, navigation safety, environmental protection, oil, gas, and mineral production, and marine bioresources harvesting.

## 6. Conclusions and Policy Implications

In this study, the EBM-DEA model is used to analyze the eco-efficiency of the 28 Russian cities along the NSR for 10 years from 2010 to 2019. This is a meaningful attempt to study the eco-efficiency of these Russian cities through quantitative analysis. This study expands the traditional TFEE model to make it suitable for the evaluation of eco-efficiency of these Russian cities. Compared with the conventional TFEE models, more pollutant variables (negative outputs) are added to the model in this study. Regarding the policy implications, in order to achieve higher eco-efficiency, the related authorities and stakeholders should devote more resources in multiple fields to the cities along the eastern and western section of the NSR. They need to develop renewable energy or eco-efficient projects such as using wind and geothermal energy resources in these cities. It is necessary to develop more industrial sectors (e.g., fishery, shipbuilding, and tourism) that are not limited to traditional energy industries based on the regional characteristics in these cities, and extend economic functions of these cities. Investment support tools should be used to attract investment to these Russian cities. It is important to renovate obsolete infrastructures, and carry out more regional infrastructure and transportation projects. It is also suggested to promote the further construction and development of the projects about oil, gas, and mineral extraction in the Russian Arctic. Particular emphasis should be placed on the circular economy for the sustainable development of the Russian Arctic cities. The clean-up of environmental damage and establishment of specially protected natural territories and reserves are effective measures for the balance between economic development and environmental protection. More resources should be invested to the related research for the sustainable development of the Russian cities along the NSR.

However, there are still unconsidered issues in this study, which can be further studied. There are many variables in the eco-efficiency model that have a carryover effect. The dynamic model will be improved to study the effect. In addition, the single process model usually ignores the conflicts between different departments within the process. The network methods will be used to conduct further research on existing models to discover the impact of conflicts between departments. Finally, the existing evaluation methods are based on the assumption of diminishing marginal productivity. It ignores that the existence of economies of scale may cause evaluation errors. Finding out how to consider the effect of economies of scale and the effect of diminishing marginal productivity within an evaluation method requires further attention.

## Figures and Tables

**Figure 1 ijerph-18-06097-f001:**
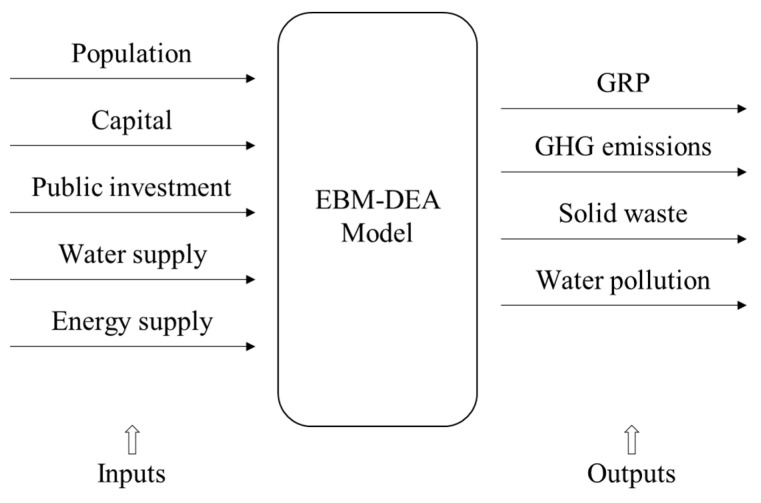
The framework of the EBM models.

**Figure 2 ijerph-18-06097-f002:**
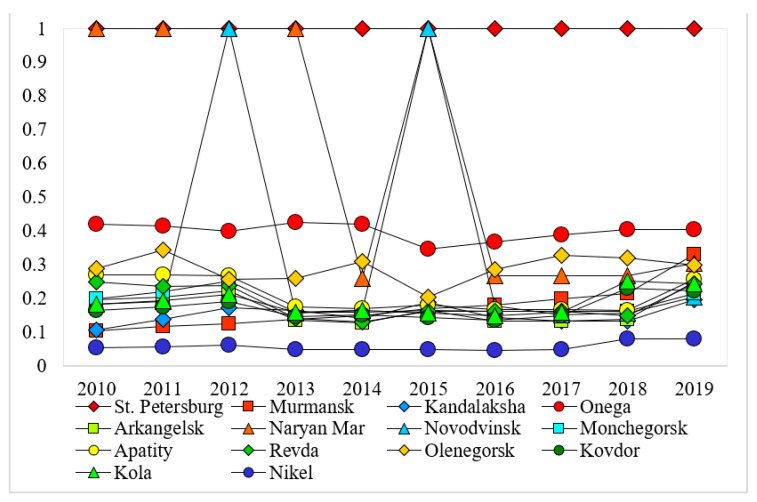
Variations in eco-efficiency scores of cities along the western section of the NSR across the years 2010–2019.

**Figure 3 ijerph-18-06097-f003:**
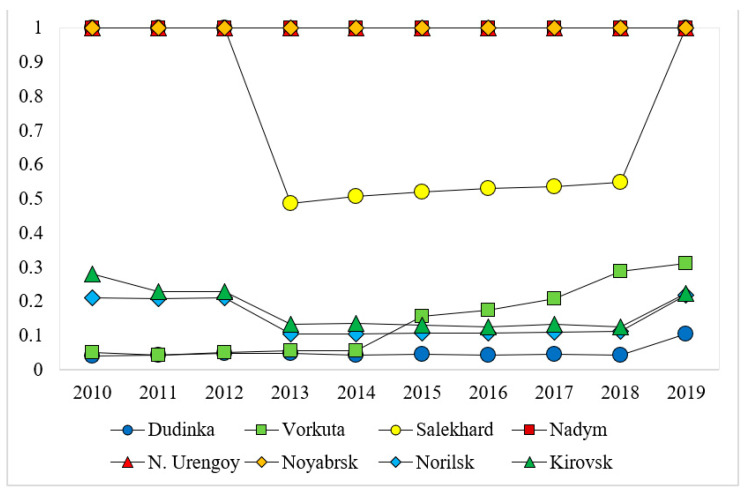
Variations in eco-efficiency scores of cities along the central section of the NSR across the years 2010–2019.

**Figure 4 ijerph-18-06097-f004:**
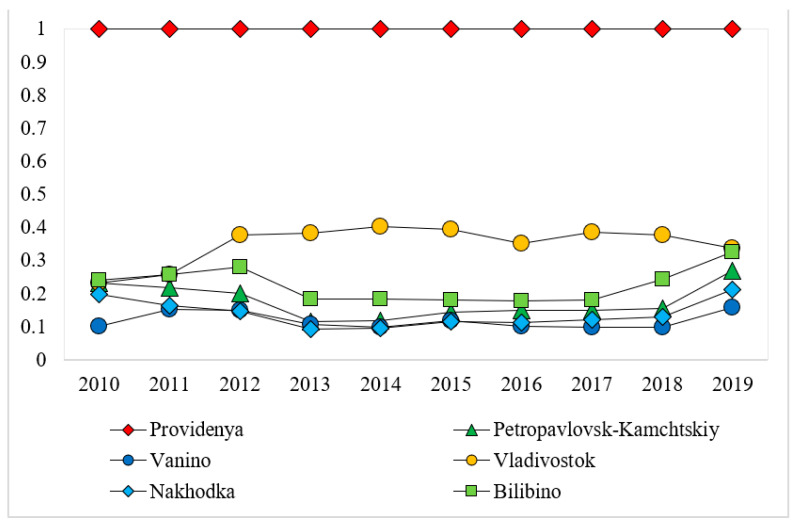
Variations in eco-efficiency scores of cities along the eastern section of the NSR across the years 2010–2019.

**Figure 5 ijerph-18-06097-f005:**
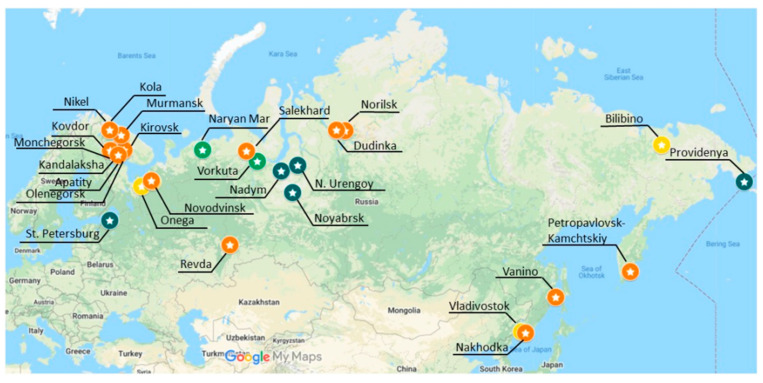
Spatial distribution of eco-efficiency of the cities along the NSR. Note: the dark green, green, yellow, and orange markers in Figure 5 indicate that the associated cities are fully, highly, averagely, and poorly eco-efficient, respectively.

**Figure 6 ijerph-18-06097-f006:**
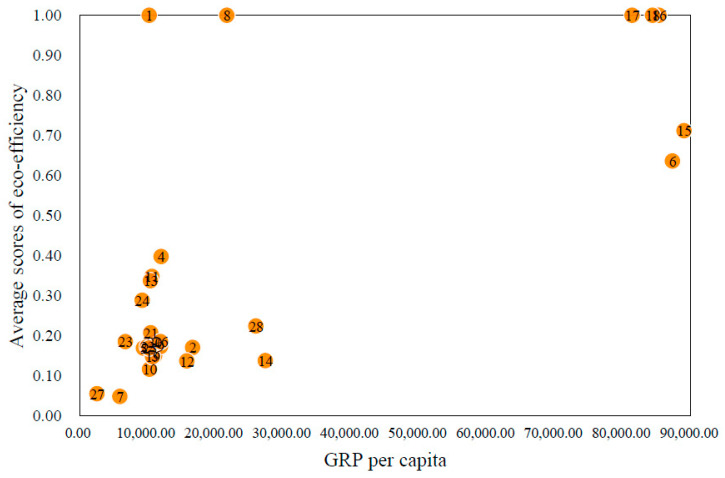
The relationship between GRP per capita and eco-efficiency of the cities along the NSR. Note: the number on the orange point in Figure 6 are the numbers in the “No.” column of Table 3, which represent the respective cities.

**Table 1 ijerph-18-06097-t001:** Units and explanations of the variables used in the models.

Variables	Units and Explanations
Population	Number of people (in million)
Capital	Billion US Dollars
Public investment	Billion US Dollars
Water supply	Average daily consumption (in thousand m^3^)
Energy supply	Specific fuel consumption for electric power generation by thermal power plants (in grams of conventional fuel/kWh)
GRP	Current prices (in million US Dollars)
GHG emissions	Thousand tons of CO_2_ equivalent
Solid waste	Thousand tons
Water pollution	Million m^3^

Source: Official Statistics of Russian Federal State Statistic Service [52].

**Table 2 ijerph-18-06097-t002:** Descriptive statistics of the inputs and outputs of models.

Year	Items	Population	Capital	Public Investment	Water Supply	Energy Supply	GRP	GHG Emissions	Solid Waste	Water Pollution
2010	Mean	0.294	0.906	0.285	209.050	160.879	1801.840	753.920	113.689	223.272
	St. dev	0.918	2.942	1.008	389.431	131.199	4813.050	3781.178	303.325	249.335
	Min	0.002	0.000	0.000	3.642	4.548	12.153	0.431	2.423	10.556
	Max	4.879	15.450	5.169	1994.700	410.540	25,632.220	20,045.000	1635.000	1105.000
2011	Mean	0.294	0.972	0.312	138.722	158.897	2051.199	705.959	125.630	215.371
	St. dev	0.921	3.098	1.076	169.307	129.678	5178.155	3533.642	311.461	243.929
	Min	0.002	0.000	0.000	3.690	4.487	13.838	0.530	3.118	9.408
	Max	4.899	16.310	5.477	490.095	406.210	27,391.429	18,734.000	1682.000	1099.000
2012	Mean	0.296	1.051	0.331	195.148	157.581	2305.258	675.684	136.720	210.040
	St. dev	0.931	3.351	1.001	361.947	128.776	5770.637	3392.140	318.662	239.812
	Min	0.002	0.000	0.000	3.925	4.440	15.280	0.592	3.798	9.954
	Max	4.953	17.661	5.031	1853.300	402.880	30,541.429	17,982.000	1723.000	1089.000
2013	Mean	0.299	1.170	0.495	190.811	156.566	2526.736	515.874	148.333	200.559
	St. dev	0.945	3.770	1.426	353.281	127.954	6212.969	2551.451	325.078	232.282
	Min	0.002	0.000	0.000	4.074	4.417	16.608	0.684	4.500	9.408
	Max	5.028	19.847	6.787	1808.800	401.220	32,821.110	13,532.000	1758.000	1071.000
2014	Mean	0.302	1.203	0.535	184.657	155.318	2639.384	444.397	157.089	195.751
	St. dev	0.964	3.782	1.485	351.059	127.038	6231.924	2171.446	325.397	229.917
	Min	0.002	0.000	0.000	4.050	4.376	18.333	0.763	5.114	9.226
	Max	5.131	19.847	6.787	1808.800	398.310	32,821.110	11,521.000	1758.000	1071.000
2015	Mean	0.304	1.025	0.529	173.531	153.106	3171.395	316.194	168.175	189.985
	St. dev	0.975	2.808	1.468	319.762	125.301	8124.212	1512.076	339.041	222.252
	Min	0.002	0.000	0.000	4.027	4.322	20.273	0.862	5.619	9.002
	Max	5.191	14.406	6.906	1635.600	394.450	43,199.600	8028.000	1831.000	1032.000
2016	Mean	0.304	1.171	0.618	170.286	151.317	3676.450	312.232	178.955	183.956
	St. dev	0.981	3.348	1.934	313.670	123.977	10,017.832	1492.260	348.212	216.759
	Min	0.002	0.000	0.000	3.889	4.264	22.388	0.947	6.206	8.862
	Max	5.225	17.320	9.690	1603.900	390.320	53,459.743	7923.000	1878.000	1009.000
2017	Mean	0.306	1.248	0.635	165.552	149.893	3888.176	294.402	188.363	177.466
	St. dev	0.992	3.709	1.908	304.357	122.924	10,254.834	1404.213	357.478	211.028
	Min	0.002	0.000	0.000	3.962	4.219	24.725	1.046	6.655	8.554
	Max	5.281	19.430	9.407	1555.900	387.040	54,636.829	7456.000	1926.000	989.000
2018	Mean	0.308	1.420	0.802	110.851	149.194	4231.944	310.924	195.873	171.473
	St. dev	1.005	4.103	2.280	133.938	122.405	11,228.630	1494.388	366.872	205.253
	Min	0.002	0.000	0.000	4.018	4.196	26.404	1.117	6.923	8.386
	Max	5.351	21.210	10.677	396.330	385.450	59,907.000	7933.000	1976.000	963.000
2019	Mean	0.309	1.594	0.957	159.041	148.700	4183.949	296.937	205.023	167.874
	St. dev	1.011	3.761	2.594	291.508	122.038	10,395.243	1425.171	376.476	202.240
	Min	0.002	0.000	0.000	4.091	4.181	28.932	1.065	7.297	8.274
	Max	5.383	17.910	11.359	1489.700	384.340	55,285.710	7566.000	2026.000	951.000

**Table 3 ijerph-18-06097-t003:** Eco-efficiency scores of the cities along the NSR for 2010–2019.

No.	Cities	Eco-Efficiency Scores									
		2010	2011	2012	2013	2014	2015	2016	2017	2018	2019
1	St. Petersburg	1	1	1	1	1	1	1	1	1	1
2	Murmansk	0.102	0.115	0.125	0.136	0.148	0.169	0.178	0.197	0.213	0.329
3	Kandalaksha	0.106	0.136	0.172	0.164	0.139	0.188	0.143	0.131	0.131	0.194
4	Onega	0.420	0.414	0.397	0.426	0.419	0.345	0.366	0.386	0.403	0.403
5	Arkangelsk	0.196	0.204	0.221	0.135	0.128	0.164	0.132	0.132	0.136	0.249
6	Naryan-Mar	1	1	1	1	0.258	1	0.266	0.267	0.268	0.305
7	Dudinka	0.040	0.041	0.047	0.046	0.041	0.045	0.042	0.044	0.043	0.103
8	Provideniya	1	1	1	1	1	1	1	1	1	1
9	Petropavlovsk-Kamchatskiy	0.230	0.217	0.200	0.113	0.118	0.144	0.149	0.148	0.155	0.267
10	Vanino	0.099	0.152	0.149	0.105	0.098	0.117	0.101	0.097	0.098	0.156
11	Vladivostok	0.231	0.258	0.376	0.381	0.401	0.392	0.351	0.386	0.375	0.336
12	Nakhodka	0.195	0.161	0.146	0.091	0.095	0.114	0.112	0.120	0.128	0.212
13	Novodvinsk	0.183	0.190	1	0.157	0.164	1	0.177	0.149	0.155	0.202
14	Vorkuta	0.049	0.041	0.050	0.055	0.056	0.154	0.174	0.207	0.288	0.311
15	Salekhard	1	1	1	0.485	0.506	0.520	0.530	0.536	0.547	1
16	Nadym	1	1	1	1	1	1	1	1	1	1
17	N. Urengoy	1	1	1	1	1	1	1	1	1	1
18	Noyabrsk	1	1	1	1	1	1	1	1	1	1
19	Norilsk	0.211	0.206	0.210	0.105	0.104	0.106	0.106	0.110	0.111	0.218
20	Monchegorsk	0.197	0.218	0.250	0.158	0.159	0.163	0.161	0.161	0.161	0.213
21	Apatity	0.270	0.270	0.266	0.175	0.169	0.179	0.167	0.165	0.164	0.256
22	Kirovsk	0.280	0.228	0.229	0.133	0.136	0.129	0.125	0.133	0.124	0.223
23	Revda	0.249	0.234	0.236	0.141	0.129	0.160	0.160	0.160	0.148	0.242
24	Olenegorsk	0.288	0.342	0.256	0.260	0.309	0.203	0.285	0.328	0.320	0.299
25	Kovdor	0.163	0.173	0.189	0.145	0.151	0.143	0.136	0.147	0.229	0.222
26	Kola	0.181	0.193	0.210	0.157	0.163	0.155	0.148	0.158	0.251	0.241
27	Nikel	0.052	0.055	0.062	0.047	0.048	0.047	0.045	0.048	0.080	0.078
28	Bilibino	0.239	0.257	0.279	0.183	0.183	0.180	0.178	0.181	0.243	0.324

**Table 4 ijerph-18-06097-t004:** The average annual percentage change of input and output variables for the Russian cities along the NSR.

Cities/DMUs	S-					S+	SB		
	Population	Capital	Public Investment	Water Supply	Energy Supply	GRP	GHG	Solid Waste	Water Pollution
St. Petersburg	0.00%	0.00%	0.00%	0.00%	0.00%	0.00%	0.00%	0.00%	0.00%
Murmansk	36.78%	97.43%	89.19%	51.22%	40.77%	279.62%	1.40%	61.16%	87.46%
Kandalaksha	18.27%	96.77%	52.80%	57.19%	48.02%	615.46%	21.79%	74.23%	90.39%
Onega	67.39%	26.71%	9.56%	14.16%	0.00%	140.99%	91.71%	52.20%	84.00%
Arkangelsk	0.00%	62.96%	55.59%	27.39%	12.14%	968.55%	93.62%	62.55%	87.27%
Naryan-Mar	0.00%	47.67%	49.81%	21.31%	15.16%	1.73%	42.98%	46.91%	48.82%
Dudinka	7.35%	95.28%	87.83%	93.51%	92.12%	1424.91%	22.77%	75.00%	90.97%
Provideniya	0.00%	0.00%	0.00%	0.00%	0.00%	0.00%	0.00%	0.00%	0.00%
Petropavlovsk-Kamchatskiy	0.00%	71.02%	58.98%	36.05%	17.98%	827.29%	99.95%	50.01%	85.68%
Vanino	1.21%	61.86%	99.33%	69.08%	62.45%	783.20%	40.36%	83.88%	94.41%
Vladivostok	40.59%	80.15%	0.00%	31.33%	0.00%	298.65%	46.58%	38.82%	30.74%
Nakhodka	0.00%	98.68%	57.71%	51.55%	55.18%	766.08%	94.24%	65.87%	88.51%
Novodvinsk	8.47%	79.26%	30.16%	30.30%	19.65%	534.39%	62.55%	74.93%	78.08%
Vorkuta	12.90%	99.22%	98.46%	30.00%	15.00%	1960.49%	30.00%	15.00%	0.00%
Salekhard	0.00%	59.71%	59.62%	5.87%	5.87%	2.13%	5.87%	5.87%	5.87%
Nadym	0.00%	0.00%	0.00%	0.00%	0.00%	0.00%	0.00%	0.00%	0.00%
N. Urengoy	0.00%	0.00%	0.00%	0.00%	0.00%	0.00%	0.00%	0.00%	0.00%
Noyabrsk	0.00%	0.00%	0.00%	0.00%	0.00%	0.00%	0.00%	0.00%	0.00%
Norilsk	0.00%	59.79%	69.78%	48.35%	52.24%	807.67%	93.86%	63.31%	87.77%
Monchegorsk	0.00%	72.53%	86.16%	7.12%	7.12%	768.69%	87.88%	71.16%	90.34%
Apatity	0.00%	58.96%	77.69%	4.02%	4.02%	722.55%	89.98%	70.41%	90.01%
Kirovsk	0.00%	65.12%	83.10%	20.53%	20.63%	824.91%	84.45%	60.62%	85.35%
Revda	0.00%	63.99%	56.05%	5.95%	5.95%	1175.02%	5.95%	5.95%	5.95%
Olenegorsk	64.48%	80.72%	0.00%	46.83%	15.08%	223.27%	67.48%	60.79%	53.73%
Kovdor	0.00%	33.45%	69.29%	0.00%	80.72%	860.22%	80.17%	77.35%	78.94%
Kola	0.00%	37.03%	69.71%	0.00%	82.24%	699.84%	82.19%	79.29%	81.01%
Nikel	0.00%	38.98%	69.87%	0.00%	83.29%	3764.85%	83.58%	80.63%	82.43%
Bilibino	0.00%	62.10%	77.79%	0.00%	91.20%	282.10%	94.09%	90.72%	93.16%

## Data Availability

The data that support the findings of this study are openly available in official statistics of Russian Federal State Statistic Service at https://eng.gks.ru/folder/11335.

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
