# Peer review of "Eco-Efficiency Analysis for the Russian Cities along the Northern Sea Route: A Data Envelopment Analysis Approach Using an Epsilon-Based Measure Model"

_ijerph, 2021, doi:10.3390/ijerph18116097_

Round 1

Reviewer 1 Report

Global review comments

a) Tables/Figures formatting must be improved.

b) The authors must be explain how obtained the data information.

Other comments

Line 154: All acronyms must be clarify in text

Line 225: Section 3 should be improved

Line 240: Equation (2): all variable must be clarified

Line 283: Table 2 must be represented using one page

Line 338: Please clarify “It is clear from Table 4 that the efficiency of capital and public investment of many Russian Arctic cities is relatively low, as 18 out of 28 cities need to have a more-than-50% reduction in both capital and public investment to achieve eco-efficiency.”

Line 439: The Supplementary Materials are not available.

Line 456: Please clarify “The data presented in this study is contained within the article.”

Reviewer 2 Report

Comments are located in the attached file 

Reviewer 3 Report

1. The work is exciting and essential to the field.
    2. I have minor considerations to make:
        a. Rewrite the first person pronouns into third person (impersonal form).
        b. Revise reference #35. Consider that there are more recent publications.
        c. Specify the reason for the temporal cut-off being only up to the year 2019.
